# A Bischler-Napieralski and homo-Mannich sequence enables diversified syntheses of sarpagine alkaloids and analogues

Hanyue Qiu[1,2], Xinghai Fei[1,2], Jiaojiao Yang[1,2], Zhen Qiao[1,2], Shan Yuan[1,2], Hu Zhang[1], Ling He[1] & Min Zhang [1] ✉

Sarpagine alkaloids offer signicant opportunities in drug discovery, yet the efficient total syntheses and diverse structural modifications of these natural products remain highly challenging due to the architectural complexity. Here we show a homo-Mannich reaction of cyclopropanol with imines generated via a Bischler-Napieralski reaction enables a protecting-group-free, redox economic, four-step access to the tetracyclic sarpagine core from L-tryptophan esters. Based on this advancement, diversified syntheses of sarpagine alkaloids and analogues are achieved in a short synthetic route. The systematic anticancer evaluation indicates that natural products vellosimine and $N_a$-methyl vellosimine possess modest anticancer activity. Intensive structural optimization of these lead molecules and exploration of the structure−activity relationship lead to the identification of analogue **15ai** with an allene unit showing a tenfold improvement in anticancer activities. Further mechanism studies indicate compound **15ai** exertes antiproliferation effects by inducing ferroptosis, which is an appealing non-apoptotic cell death form that may provide new solutions in future cancer therapies.

The Mannich reaction, typically involving the reaction of an enol with an imine or an iminium ion, is a well-established method for the synthesis of β-amino carbonyl compounds (Fig. 1a), while the analogous reactions of homo-enol or its equivalents with imines or iminium ions are much less explored[1,2]. As one of the most common homo-enol equivalents, cyclopropanol can undergo ring-opening additions to various carbonyl compounds, however, its reaction with iminium ions or imines to yield γ-amino carbonyl products remains scarce (Fig. 1a)[3–16]. Recently, we reported the first homo-Mannich reaction of cyclopropanol with the iminium ion generated by C−H oxidation, which provided rapid access to the azabicyclo[3.3.1]nonane core and thus culminated in efficient syntheses of sarpagine alkaloids[17,18]. The major drawback of this method is the use of the N-PMP group, which brought about three unrewarding and low-yielding transformations for its introduction and removal. To circumvent this issue, we

envisioned that the Bischler-Napieralski reaction of formamide (**1**) could afford an iminium ion (**2**), which could also theoretically serve as a homo-Mannich reaction partner with the cyclopropanol moiety (Fig. 1b). If this strategy could be successfully implemented, a protecting group-free, redox-economic, concise access to the sarpagine core indole-fused 9-azabicyclo[3.3.1]nonane (**3**) could be expected. This success would allow a further shortened synthetic route to the sarpagine alkaloids (selected members are shown in Fig. 1c).

Sarpagine alkaloids represent one of the most prominent families of monoterpene indole alkaloids, with >100 members having been isolated from plants with medicinal significance[19–21]. Over decades of intensive synthetic studies, seminal and effective methods have been developed to construct the 9-azabicyclo[3.3.1]nonane central core, including Dieckmann cyclization[22–29], olefin metathesis[30], indolyl Friedel-Crafts reaction[31], aza-Achmatowicz/indole cyclization[32], [5 + 2]-

[1]Chongqing Key Laboratory of Natural Product Synthesis and Drug Research, Innovative Drug Research Center, School of Pharmaceutical Sciences, Chongqing University, Chongqing 401331, China. [2]These authors contributed equally: Hanyue Qiu, Xinghai Fei, Jiaojiao Yang, Zhen Qiao, Shan Yuan. ✉e-mail: minzhang@cqu.edu.cn

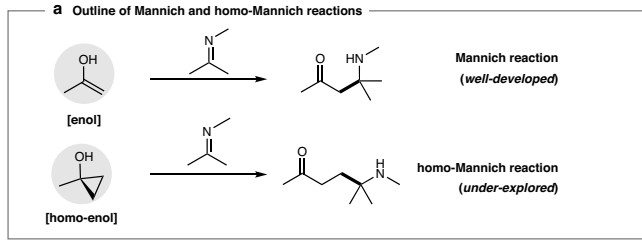

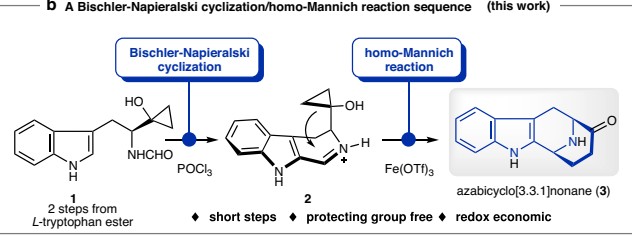

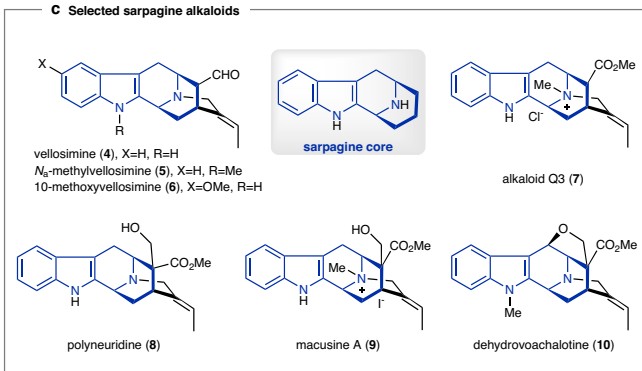

**Fig. 1 | Background and study synopsis. a** Outline of the Mannich and homo-Mannich reaction. **b** Assembly of the azabicyclo[3.3.1]nonane skeleton via a sequential Bischler-Napieralski cyclization/homo-Mannich reaction. **c** The sarpagine alkaloids targeted in this study.

cycloaddition/ring enlargement[33], Fischer indolization[34], Mannich-type cyclization[35,36], and amide-alkene cyclization[37], which led to many elegant syntheses of sarpagine alkaloids[17–21,38,39]. Despite these advances, only a small proportion of sarpagine alkaloids have been subjected to preliminary tests including anti-hypertension, anti-cancer, and anti-inflammation[19–21], and in-depth biological evaluations remained to be performed. Therefore, the development of new and efficient methods, which would allow rapid access to multiple natural products and be adaptable to syntheses of structurally diverse analogues, is of high value for both synthetic chemistry and medicinal chemistry. We report here a Bischler-Napieralski/homo-Mannich sequence for the construction of sarpagine core (**3**), which enables diversified syntheses and systematic anticancer evaluation of sarpagine alkaloids and analogues.

The abovementioned homo-Mannich strategy has several potential strengths. First, chiral cyclopropanol **1** with various substitution patterns can be conveniently prepared in two steps via *N*-formylation and Kulinkovich reaction starting from the readily available *L*-tryptophan esters, which would tremendously facilitate both the natural product synthesis and analogue preparation[4]. Second, Bischler-Napieralski cyclization of formamide onto the indole ring to generate an imine is well-documented[40,41]. Most importantly, our experiences in homo-Mannich reaction of cyclopropanol indicate the feasibility of the crucial cyclization step[17,18]. However, several concerns remained to be addressed. First, the Bischler-Napieralski reaction is usually carried out with POCl₃ or Tf₂O[40,41], which are strongly acidic and thus may not tolerate the acid-labile cyclopropanol moiety.

Second, our previous successes in homo-Mannich reaction of cyclopropanol were limited to the reaction with *N*-aryl iminium ions, of which the aryl group facilitates the reaction by stabilizing the generated nitrogenous radical intermediate, and the reaction with imines or iminium ions without a *N*-aryl substituent is beyond our exploration scope[17,18]. A solution to these concerns is to find appropriate reaction conditions for the Bischler-Napieralski reaction without interfering with the labile cyclopropanol unit, and, more importantly, to achieve homo-Mannich reaction of cyclopropanol with a new iminium partner.

## Results and discussion

At the outset of this study, various reaction parameters for the Bischler-Napieralski reaction of **1**, including amide activators (e.g., Tf₂O, POCl₃, and P₂O₅), bases (e.g., 2-ClPyr, and DTBMP), and solvents, were systematically screened[41,42]. Eventually, the use of POCl₃ with THF as the solvent was determined to be the most suitable, which showed good tolerance for the cyclopropanol moiety while maintaining reasonable reactivity. Under this condition, a series of imines with electronically different substituents including F, Cl, Br, and MeO at various positions of the indole ring were successfully prepared in the form of hydrochloride salts as indicated by the ion chromatography (Figs. 2, 2a, 2e–2l). Next, various metal salts (e.g., Cu, Mn, Fe, Sc, Zn, and Ag) were screened for the subsequent homo-Mannich reaction (See Supplementary Information for details). Among them, Fe(OTf)₃ was identified as the best metal salt, providing a series of **3** in 51–76% yields (Figs. 2, 3a, 3e–3l), while CuCl₂ used for the *N*-aryl iminium ions in our previous works produced none of the desired products[17,18]. Under the optimal conditions, substrates **2b–2d** in a free imine form also readily participated in this reaction, furnishing **3b–3d** with *N*-protecting groups that can be removed under different conditions (Fig. 2). Cyclopropanol is prone to be oxidized by Fe(III) salts to a β-keto radical intermediate[3–7], therefore, this homo-Mannich reaction was postulated to go through a radical process.

Having established a new method for the construction of 9-azabicyclo[3.3.1]nonane core, grams of **3a** and **3l** were prepared from *L*-tryptophan esters **11a** and **11l** via a four-step sequence including *N*-formylation, Kulinkovich cyclopropanation, Bischler-Napieralski cyclization, and homo-Mannich reaction (Fig. 3a). Propargylation of **3a** and **3l** with **12** and then ketone α-allenylation secured the pentacyclic sarpagine core with an allene group installed (**14a, 14l**). Wittig reaction of **14a** and **14l** followed by selective hydrogenation of the allene group from the less hindered face furnished vellosimine (**4**) and 10-methoxyvellosimine (**6**) in eight total steps from *L*-tryptophan ester **11a** and **11l**, respectively[43,44]. The total synthesis of *N*ₐ-methylvellosimine (**5**) was also accomplished through the same reaction sequence after *N*-methylation of **14a**. The total synthesis of alkaloids Q3 (**7**) was then carried out (Fig. 3b). Using **15a** as the branch point, oxidation of aldehyde with NIS/MeOH gave ester **16**. Partial saturation of the allene unit of **16** provided **17**, which was transformed to alkaloids Q3 (**7**) via sequential *N*-quaternization and anion exchange following Cook's protocol[23,24]. Starting with **15a**, a sequence including *N*-tosylation, domino aldol/Cannizzaro reaction by treating with Cs₂CO₃/paraformaldehyde[27,45], and *N*-detosylation produced diol **18** in high efficiency. Oxidation of **18** with DDQ yielded ether **19**, which was transformed to ester **20** by AZADO-catalyzed oxidation and then treatment of NIS/MeOH. Release of the hydroxyl group via TFA/Et₃SiH and partial saturation of the allene group furnished polyneuridine (**8**), which could be transformed to macusine A (**9**) through *N*-quaternization[26,27]. The total synthesis of dehydrovoachalotine (**10**) was accomplished via a two-step sequence including *N*-methylation and then allene hydrogenation using the shared intermediate **20**.

With substantial quantities and structural diversity of naturally occurring sarpagine alkaloids in hand, the stage was set for systematic biological studies. Preliminarily, the collection of natural sarpagine alkaloids was subjected to the in vitro cancer cell assays,

**Fig. 2 | Substrate scope.** Conditions: **a 1** (1 mmol), POCl$_3$ (5 mmol), THF (1 mL), 0 °C to room temperature, 5–20 min; **b 2** (0.2 mmol), Fe(OTf)$_3$ (0.4 mmol), dioxane (2 mL), room temperature, 15–30 min. Yields are for the isolated **3** and the ones in parentheses are for **2. 2b, 2c**, and **2d** were prepared by *N*-protection of **1a** and used as the free imine form.

and it turned out that vellosimine (**4**) and $N_a$-methylvellosimine (**5**) exhibited moderate antiproliferation activity (Table 1). Based on this finding, we designed and synthesized more diversified derivatives to map out the structure-activity relationships (SARs). The bulky *p*-toluenesulfonyl (Ts) group at the $N_a$ site and the allene motif were found to be key to bioactivity improvement (**15ad**, MDA-MB-231 IC$_{50}$ = 3.07 μM). Furthermore, the chlorine atom at the 5-position of the scaffold slightly elevated the potency (**15ai**, MDA-MB-231 IC$_{50}$ = 2.03 μM), whilst other groups at the same position, or halogen atoms at other positions (**15ae**–**15ag**, **15aj**, **15al**), generally attenuated the potency. The replacement of the Ts group with a 4-(trifluoromethyl)benzene sulfonyl group annihilated the antiproliferation activity (**15am**, IC$_{50}$ > 50 μM). Besides, the aldehyde group was found vital to maintain the antiproliferation activity[46]. Its oxidative or reductive products (**15an** and **15ao**), or its bioisosteres (**15ap** and **15aq**), led to a complete or partial loss in activity, respectively. Finally, the partial hydrogenation of the allene did not improve the potency (**15ar**). The overall SAR study indicated that the $N_a$-Ts and the aldehyde groups are vital pharmacophore elements, and the 5-chloride along with the allene substitution favors the biological effects. Meanwhile, none of the tested compounds showed obvious cytotoxic effects on lung epithelial BEAS-2B cells (CC$_{50}$ > 100 μM) (Table 1). As compound **15ai** exhibited tenfold-improved anticancer activity compared to vellosimine (**4**), we further carried out investigations on its mechanism of action.

In brief, the flow cytometry assays indicated that **15ai** exerted a non-apoptotic cell death form, since the addition of apoptosis inhibitor Z-VAD-FMK (Z-VAD) or necroptosis inhibitor necrostatin-1 (Necro-1) could not reverse cell death of MDA-MB-231 cells treated by **15ai** (Fig. 4a, Supplementary Fig. 1). In contrast, the co-existence of ferroptosis inhibitor ferrostatin-1 (Fer-1) or ROS inhibitor *N*-acetylcysteine (NAC) remarkably rescued cell viability (Fig. 4b, Supplementary Fig. 2). Further, **15ai** has proved to induce lipid ROS and cytosolic ROS accumulation significantly, which could be suppressed by Fer-1 (Fig. 4c). The morphological analysis of the TEM (transmission electron microscopy) images also revealed that mitochondrial swelling occurred after the treatment of **15ai**, a distinguishing feature of ferroptosis (Fig. 4d)[47,48]. Collectively, these results demonstrated that **15ai** is a potent and specific ferroptosis inducer. The following evaluations on the ferroptosis system x$_c$-GSH-GPX4 pathway revealed that **15ai** dose-dependently decreased the system x$_c$-

component SLC7A11 and GSH levels, whilst did not affect the protein level of GPX4 (Fig. 4e, Fig. 4f). Hence, **15ai** was proved to downregulate SLC7A11, reduce GSH biosynthesis, trigger ROS accumulation, and induce ferroptosis in MDA-MB-231 cells. As triggering ferroptosis illuminates an appealing potential for cancer therapy, particularly for aggressive malignancies resistant to traditional therapies[49], this chemotype may be an ideal starting point for novel drug development in future studies.

In conclusion, we have developed a Bischler-Napieralski cyclization/homo-Mannich reaction sequence for the efficient construction of sarpagine core. Owing to the high efficiency of this newly developed method, both natural products and their analogues can be synthesized in a short synthetic route. The natural products vellosimine and $N_a$-methylvellosimine were found to show modest anticancer activity, and the SAR study led to the identification of analogue **15ai** with a tenfold improvement in anticancer activities. Mechanism studies indicated **15ai** exerted antiproliferation effects by inducing ferroptosis, which holds the promise in overcoming the apoptotic resistance. This study not only enriches the homo-Mannich reaction but may provide clues for the exploration of new therapeutic agents in the treatment of refractory cancer types.

## Methods
### General procedure for preparation of 3
To a solution of **2** (0.2 mmol) in dry 1,4-dioxane (2 mL) under an atmosphere of argon was added Fe(OTf)$_3$ (0.4 mmol). The reaction mixture was stirred for 10 min at room temperature, and then was quenched with a saturated aqueous solution of NaHCO$_3$ (10 mL). After filtration through a pad of Celite, the resulting mixture was extracted with EtOAc (3 × 20 mL). The combined organic phase was washed with brine (3 × 20 mL), dried over Na$_2$SO$_4$, filtered, and concentrated. The crude product was purified by flash column chromatography on silica gel to afford the pure product **3**.

### Cell culture
All cell lines were cultured at 37 °C with 5% CO$_2$ in a humidified incubator. HeLa cells were grown in MEM (Gibco) supplemented with 10% fetal bovine serum (FBS, Hyclone) and 1% penicillin/streptomycin (Gibco). MIA PaCa-2, MDA-MB-231, A549, MCF-7, A375 and BEAS-2B cells were grown in DMEM (Gibco) supplemented with 10% FBS and 1% penicillin/streptomycin. CT26, and HCT116 cells were grown in RPMI-

**Fig. 3 | Reactions and conditions. a** HCOOEt, 60 °C, 10 h; **b** TiCl$_2$(O$^i$Pr)$_2$, THF, 0 °C to rt, 3 h, **1a**: 65% from **11a**, **1l**: 64% from **11l**; **c** POCl$_3$, THF, 0 °C to rt, 30 min, **2a**: 72%, **2l**: 81%; **d** Fe(OTf)$_3$, 1,4-dioxane, rt, **3a**: 67%, **3l**: 55%; **e 12**, K$_2$CO$_3$, MeCN, reflux, 5 h; **13a**: 81%, **13l**: 52%; **f** AgNTf$_2$, pyrrolidine, 90 °C, 20 min, **14a**: 78%, **14l**: 55%; **g** MeI, KOH, TBAB, THF, 2 h, 70%; **h** $^n$BuLi, MOMPPh$_3$•Cl, THF, −78 °C, 2 h, then HCl (2 N), **15a**: 50%, **15l**: 80%, **15aa**: 73%; **i** Pd/C, H$_2$, MeOH, −30 °C, 45 min to 3 h, vellosimine (**4**): 85%, E/Z > 20:1; $N_a$-methylvellosimine (**5**): 89%, E/Z > 20:1; 10-methoxyvellosimine (**6**): 57%, E/Z > 20:1; **j** NIS, K$_2$CO$_3$, MeOH, 2 h, 85%; **k** Pd/C, H$_2$, MeOH, −40 °C, 2 h, 90%, E/Z > 20:1; **l** MeI, THF, 0 °C, 3 h; **m** AgCl, MeOH, rt, 2 d, 2 steps for 72%; **n** NaH, TsCl, DMF, 0 °C to rt, 8 h, 85%; **o** (CHO)$_n$, Cs$_2$CO$_3$, THF, 0 °C, 81%; **p** Mg, NH$_4$Cl, MeOH/benzene (1:1), rt, 3 h, 92%; **q** DDQ, THF, −78 °C, 1 h, 81%; **r** AZADO, bpy, CuCl, DMAP, air, rt, 30 min; **s** NIS, K$_2$CO$_3$, MeOH, 0 °C, 2 h, 2 steps for 79%; **t** TFA, Et$_3$SiH, CH$_2$Cl$_2$, reflux, 48 h, 69%; **u** Pd/C, H$_2$, MeOH, −30 °C, 2 h, 88%, E/Z > 20:1; **v** MeI, THF, 0 °C, 10 h, 84%; **w** MeI, KOH, TBAB, THF, rt, 1 h, 73%; **x** Pd/C, H$_2$, MeOH, −20 °C, 4 h, 83%, E/Z > 20:1.

1640 supplemented with 10% FBS and 1% penicillin/streptomycin. All cell lines were purchased from the National Collection of Authenticated Cell Cultures (China).

## Cell viability assay

The antiproliferative activities of the prepared compounds against tested cell lines were evaluated using a standard (MTT)-based colorimetric assay. Cells were plated at a density of 3000 cells/well in 96-well plates and allowed to adhere overnight. Then cells were treated with varying concentrations of synthesized compounds for 72 h. At the end of treatment, 20 μL of thiazolyl blue tetrazolium bromide (MTT) solution (5 mg/mL) was added to each well. Cells were incubated for an additional 4 h at 37 °C and then the medium was removed and replaced with 200 μL of DMSO. The absorbance (OD$_{490}$) was read on a microplate reader (SpectraMax i3x, Molecular Devices, US). In all experiments, three replicate wells were used for each drug concentration. Each assay was performed at least three times. The IC$_{50}$ values were calculated by GraphPad Prism 9, which were the mean values derived from three independent experiments.

## Cell death manner assay

To investigate the mechanism of **15ai** in cancer cell proliferation inhibition, the manners of cell death were analyzed. The cell apoptosis inhibitor Z-VAD-FMK (Z-VAD), cell necrosis inhibitor Necrostatin-1 (Necro-1), ferroptosis inhibitor Ferrostatin-1 (Fer-1), and ROS inhibitor N-acetylcysteine (NAC) were purchased and stored at −20 °C. A375 cells were seeded with a density of 5000 cells/100 μL/well into 96 well plates. After 12 h, **15ai** was added at different concentrations. Following that 10 μM Z-VAD, 10 μM Nerco-1, 10 μM Fer-1, and 5 mM NAC were added and co-incubated with **15ai**, respectively, for 72 h. Then, 20 μL MTT solution (5 mg/mL) was added and incubated at 37 °C for an additional 4 h. Then, the supernatant was discarded, and 200 μL DMSO was added to dissolve the precipitate. After 15 min, the absorbance (OD$_{490}$) was read on a microplate reader (SpectraMax i3x, Austria). In all experiments, three replicate wells were used for each drug concentration. Each assay was performed at least three times.

## Cell apoptosis analysis

Approximately 10$^5$ cells/well were plated in 12-well plates. Subsequently, cells were then incubated with compound **15ai** at different

**Table 1 | Proliferation inhibitory activities of sarpagine alkaloids (4–10) and selected analogs against cancer cell lines and cytotoxicity towards normal cells[a]**

| Compound | IC$_{50}$ (μM) | | | | | | | | CC$_{50}$ (μM) |
|---|---|---|---|---|---|---|---|---|---|
| | MIA PaCa-2 | MDA-MB-231 | Hela | A549 | MCF-7 | CT26 | HCT 116 | A375 | BEAS-2B |
| vellosimine (**4**) | 32.58 | 21.15 | 24.11 | 23.23 | 39.21 | 20.2 | 27.51 | 28.59 | >100 |
| $N_a$-methylvellosimine (**5**) | 26.69 | 20.69 | 22.28 | 20.85 | 41.77 | 21.35 | 23.58 | 27.62 | >100 |
| alkaloids **6–10** | >50 | >50 | >50 | >50 | >50 | >50 | >50 | >50 | >100 |
| **15ad** | 12.14 | 3.07 | 4.37 | 10.44 | 11.26 | 8.14 | 5.25 | 3.57 | >100 |
| **15ae** | 40.27 | 25.41 | 32.07 | 42.87 | 32.11 | 30.12 | 28.01 | 20.40 | >100 |
| **15af** | 20.18 | 5.39 | 8.56 | 14.81 | 18.11 | 15.82 | 9.98 | 7.51 | >100 |
| **15ag** | 22.76 | 8.59 | 9.89 | 18.16 | 20.88 | 17.29 | 12.44 | 8.88 | >100 |
| **15ai** | 8.32 | 2.03 | 2.95 | 12.98 | 13.91 | 5.98 | 4.78 | 2.64 | >100 |
| **15aj** | 24.12 | 14.02 | 18.82 | 26.17 | 30.27 | 24.28 | 18.88 | 13.44 | >100 |
| **15al–15ap** | >50 | >50 | >50 | >50 | >50 | >50 | >50 | >50 | >100 |
| **15aq** | 20.18 | 8.12 | 10.86 | 19.16 | 24.76 | 15.72 | 10.28 | 7.48 | >100 |
| **15ar** | 13.31 | 4.63 | 6.16 | 14.08 | 17.76 | 9.39 | 6.96 | 4.54 | >100 |

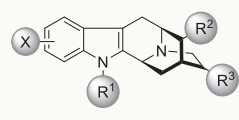

**15ad**: X=H, R¹=Ts, R²=CHO, R³= ≡·═
**15ae**: X=4-Br, R¹=Ts, R²=CHO, R³= ≡·═
**15af**: X=5-Br, R¹=Ts, R²=CHO, R³= ≡·═
**15ag**: X=6-Br, R¹=Ts, R²=CHO, R³= ≡·═
**15ai**: X=5-Cl, R¹=Ts, R²=CHO, R³= ≡·═
**15aj**: X=6-F, R¹=Ts, R²=CHO, R³= ≡·═
**15al**: X=5-OMe, R¹=Ts, R²=CHO, R³= ═

**15am**: X=H, R¹=*p*-CF₃-PhSO₂, R²=CHO, R³= ≡·═
**15an**: X=H, R¹=Ts, R²=COOH, R³= ≡·═
**15ao**: X=H, R¹=Ts, R²=CH₂OH, R³= ≡·═
**15ap**: X=H, R¹=Ts, R²= (CF₃), R³= ≡·═
**15aq**: X=H, R¹=Ts, R²= , R³= ≡·═
**15ar**: X=H, R¹=Ts, R²=CHO, R³=

[a]Data are presented as the mean values from at least three independent experiments.

concentrations for 24 h. Non-treated wells received an equivalent volume of ethanol (<0.1%). After incubation, cells were trypsinized, washed in PBS, and centrifuged at 2000 x g for 5 min. The pellet was resuspended in 500 μL staining solution (containing 5 μL AnnexinV-FITC and 5 μL PI in binding buffer), mixed gently, and incubated for 15 min at room temperature in the dark. The samples were then analyzed by the analytical flow cytometry (CytoFLEX, Becton Dickinson, US). Data analysis was performed using the FlowJo 7.6.1 software. The starting cell population gating by Forward Scatter and Side Scatter was used to make sure doublet exclusion. Only single cell was used for analysis.

## Measurement of lipid and cytosolic ROS
Lipid ROS was analyzed by flow cytometry: cells were plated at a density of $2 \times 10^5$ cells/well in 6-well plates. After treatment as indicated, cells were stained with 5 μM BODIPY-C11 dye (diluted in serum-containing DMEM) for 30 min at 37 °C in an incubator. For cytosolic ROS, cells were stained with 10 μM DCF (diluted in serum-free DMEM). Following staining, cells were pelleted and resuspended in 1 mL of PBS. Fluorescence intensity was measured by the analytical flow cytometry (CytoFLEX, Becton Dickinson, US). A minimum of 10,000 cells were analyzed for each condition. Data analysis was performed using the FlowJo 7.6.1 software. The starting cell population gating by Forward Scatter and Side Scatter was used to make sure doublet exclusion. Only single cell was used for analysis.

## Measurement of glutathione levels
Cells were plated at a density of $2 \times 10^5$ cells/well in 6-well plates and allowed to adhere overnight. After being treated as indicated, cells were collected and prepared for measurement of total intracellular glutathione (GSH + GSSG) using the assay kit (Beyotime, S0052) according to the manufacturer's instructions. Three independent biological replicates were performed for the experiments.

## Transmission electron microscopy
Cells were plated at a density of $6 \times 10^5$ cells/dish in 10 cm culture dishes and allowed to adhere overnight. Then cells were treated with DMSO or compound **15ai** for 24 h. Cells were fixed with 2.5% glutaraldehyde in 0.1 mM phosphate buffer, followed by 1% OsO₄ for 2 h. After dehydration, cells were embedded in epoxy resin. The ultrathin sections were made by an ultramicrotome and then stained with lead citrate and uranyl acetate. Images were acquired using the transmission electron microscope (Hitachi H-7650 100kv, Hitachi, Japan). Each experiment was repeated three times independently with similar results.

## Western blotting
Cells on 6-well plates were rinsed twice with cold PBS and lysed in RIPA lysis buffer (50 mM Tris (pH 7.4), 150 mM NaCl, 1% Triton X-100, 1% deoxycholicphenyl methylsulfonyl fluoride, 5.0 mM sodium pyrophosphate, 1.0 g/mL leupeptin, 0.1 mM phenylmethylsulfonyl fluoride, and 1 mM DTT) containing a protease inhibitor (PMSF) mixture at 1:100 dilution on ice for 30 min. The insoluble components of cell lysates were removed by centrifugation (4 °C, 12000 × g, 10 min), and protein concentrations were measured using a Pierce BCA protein assay kit. Proteins were separated by sodium dodecyl sulfate-polyacrylamide gel electrophoresis (SDS-PAGE) and transferred onto polyvinylidene difluoride (PVDF) membranes. Membranes were blocked using 5% skim milk in TBST and then incubated with diluted indicated primary antibodies: anti-GPX4 (1:1000 dilution, Proteintech, Cat No: 30388-1-AP), anti-SLC7A11 (1:1000 dilution, ABclonal, Cat No: A13685), anti-GAPDH (1:1000 dilution, Wanlei, Cat No: WL01114) at 4 °C with gentle shaking overnight. After washing five times (5 min each) with TBST, membranes were incubated with the HRP conjugated goat anti rabbit IgG (H + L) (1:5000 dilution, Wanlei, Cat No: WLA023) for 1 h at rt. Immunoreactive bands were visualized using an ECL detection kit (Wanlei, Cat No: WLA006) following the manufacturer's instruction on the ChemiDoc system (BioRad, Shanghai, China).

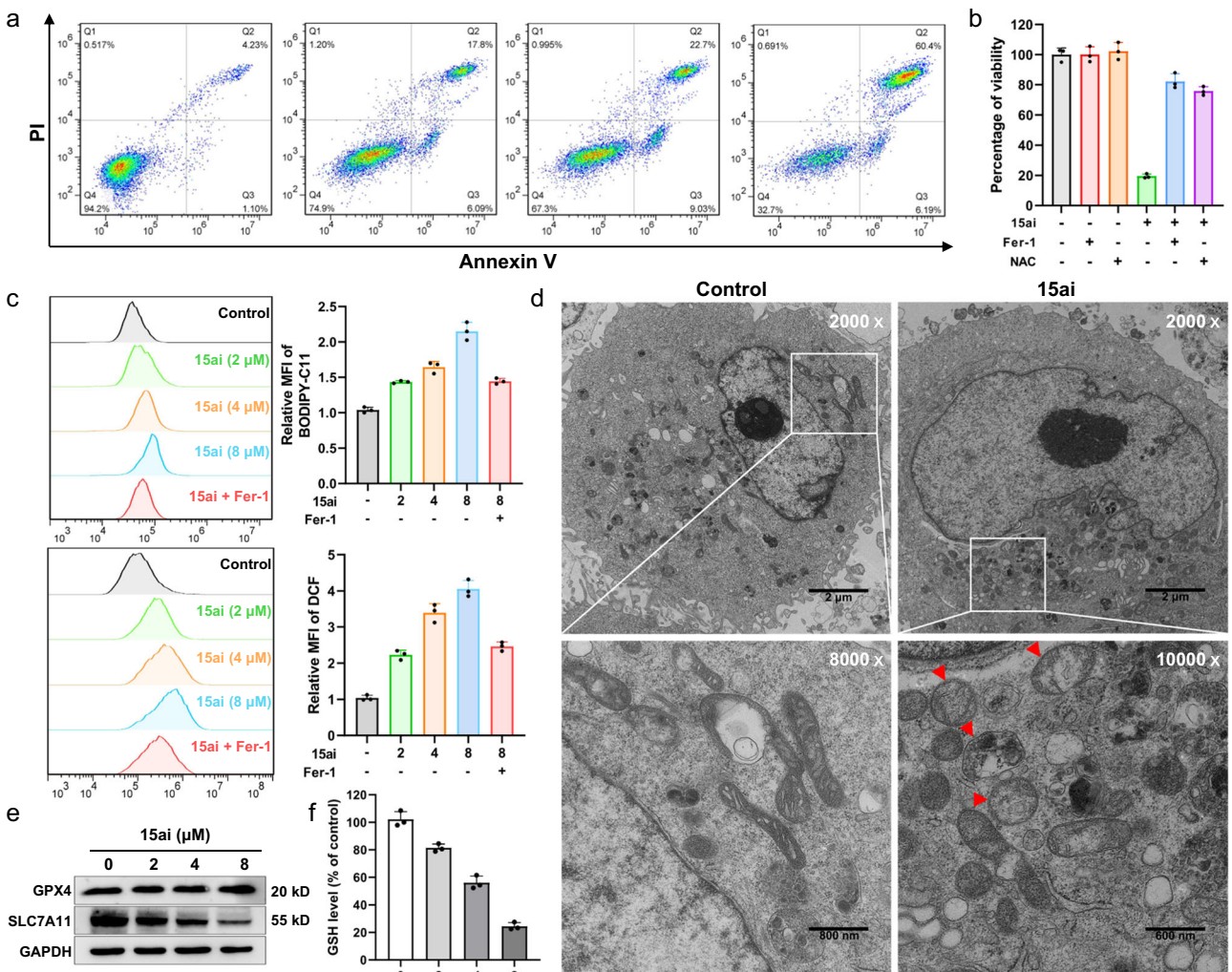

**Fig. 4 | Compound 15ai induced cell death of MDA-MB-231 cells through ferroptosis. a** Quantitative analysis of cell death induced by **15ai** (0, 2, 4 and 8 μM) was measured by Annexin V/PI staining coupled with flow cytometry. **b** The percentage of cell viability after the combination of **15ai** (10 μM) with ferroptosis inhibitor Fer-1 (10 μM) or ROS inhibitor NAC (5 mM) for 72 h. Data are presented as mean values ± SD, and *n* = 3 biologically independent replicates. **c** Cells were treated with **15ai** in the absence or presence of Fer-1 (10 μM) for 12 h. The relative lipid ROS levels were assayed via BODIPY-C11 fluorescence. The relative cytosolic ROS levels were assayed via DCFH-DA fluorescence. Data are presented as mean values ± SD, and *n* = 3 biologically independent replicates. **d** Cell morphology was observed via TEM after MDA-MB-231 cells were treated with **15ai** (4 μM) for 24 h. Each experiment was repeated three times independently with similar results. **e** Cells were treated with **15ai** (0, 2, 4 and 8 μM) for 24 h. SLC7A11 and GPX4 protein expression was measured via Western blotting. **f** The relative levels of GSH were assayed. Data are presented as mean values ± SD, and *n* = 3 biologically independent replicate.

## Reporting summary

Further information on research design is available in the Nature Portfolio Reporting Summary linked to this article.

## Data availability

All relevant data supporting the findings of this study, including experimental procedures, compound characterizations, biological activity studies are available within the Article and its Supplementary Information. The raw data of Flow Cytometry, TEM and Western Blot had been deposited in Figshare (https://doi.org/10.6084/m9.figshare.24038868). Source data are provided with this paper.

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

## Acknowledgements

This work was financially supported by the Science and Technology Innovation Key R&D Program of Chongqing (CSTB2022TIAD-STX0015), National Natural Science Foundation of China (22271033, 21922102, and 22101037), Chongqing Science and Technology Commission (CSTB2022NSCQLZX0036), Luzhou Science and Technology Program (2022-XDY-191), and Fundamental Research Funds for the Central Universities (2022CDJXY-025). We also thank Dr. Linxi Wan (SCU) for HRMS assistance.

## Author contributions

M.Z. conceived and directed the project. H.Q., X.F., J.Y., Z.Q. and S.Y. designed and performed experiments and prepared the Supplementary Information. H.Q. and S.Y. performed the bioactive investigations. H.Q., X.F., H.Z. and L.H. analyzed and interpreted the experimental data. M.Z. wrote the paper. All authors discussed the results and commented on the paper.

## Competing interests

The authors declare no competing interests.
