## [Peer Review File · Nature Communications]

A Bischler-Napieralski and Homo-Mannich Sequence Enables Diversified Syntheses of Sarpagine Alkaloids and AnaloguesReviewers' Comments:

Reviewer #1:

Remarks to the Author:

In this manuscript, Zhang and co-workers reported a highly efficient approach to access the tetracyclic core of sarpagine-type alkaloids, hinging on a sequential Bischler-Napieralski reaction and homo-Mannich reaction as the key element. While the same group has published a series of seminal works on the subject of homo-Mannich reaction of cyclopropanol with the iminium ion, the essence of the present study is to combine the unique tactics with the classical Bischler-Napieralski reaction, which offers a more redox economic method than the previous one. Based on the newly developed methodology, a collection of sarpagine-type alkaloids and their analogues have been achieved in short steps. Moreover, the authors also evaluated the anti-cancer activity of the obtained natural products and their analogs. Encouragingly, they identified one synthetic analogue that displayed significantly improved anti-cancer activity than the natural products. More importantly, the mechanism of action of the identified active compound was also conducted, which revealed that it exerted the antiproliferation effects by serving as a promoter of ferroptosis. It should be noted that ferroptosis has recently emerged as a hot subject in the biomedical research area. In this context, the present work provides a new chemotype of ferroptosis promoter, which may serve as a new lead for the development of novel anti-cancer drugs. Collectively, the reported discoveries in this work are of great significance in both chemistry and biology. The manuscript is well organized and the supporting information also looks excellent. Thus, I would like to recommend its acceptance by Nature Communication with minor revisions:

- 1) Page 1, left column, for the reference 15-35, it is better to put the corresponding reference(s) just behind the mentioned reactions. It will help the readers to know which reference(s) correlates to which reaction.
- 2) Page 2, left column, ".....to find an appropriate reaction condition for.....", should be ".....to find appropriate reaction conditions for.....".
- 3) Page 2, right column, "Among these, Fe(OTf)₃ was identified....." should be ""Among them, Fe(OTf)₃ was identified.....".
- 4) Page 2, "Starting with 15a, a sequence including N-tosylation, hydroxymethylation with Cs₂CO₃/paraformaldehyde and detosylation produced diol 18 in high efficiency." In this reaction sequence, a disproportionated reaction should occur after the hydroxymethylation reaction, since the aldehyde needs to be reduced to the alcohol. However, this reaction was not mentioned in the manuscript, which makes the sentence misleading. Moreover, the corresponding reference for the one-pot hydroxymethylation and disproportionated reaction should be included in this place.
- 5) In Fig.3, some figures look vague, which should be replaced with high-quality ones.
- 6) For the partial hydrogenation of the allene group in the total synthesis, how comes the excellent diastereoselectivity (cis vs trans)? Is there any rationalization for the result? If there are some references for this transformation, they should be included in the corresponding place.

Reviewer #2:

Remarks to the Author:

Zhang and co-workers report a Bischler-Napieralski/homo-Mannich sequence as a new entry to the core of the sarpagine alkaloids. This new synthetic approach is highly efficient, as 1) the starting materials (1) are accessible from the corresponding tryptophan derivatives, 2) the two key steps involve only very standard reagents and techniques, and 3) no protective groups are required. As a result, this new approach represents a superior strategy to access the valuable natural product scaffold 3. The authors were able to prepare not only no less than five complex alkaloids in very efficient sequences, but also a wide variety of non-natural analogues (15ad-ar), allowing for a systematic evaluation of the anticancer properties of this compound class. Intriguingly, the naturally occurring alkaloids were shown to display modest proliferation inhibition at best, however, selected synthetic analogues (e.g. 15ai) showed up to ~10-fold enhanced activity. Moreover, the authors

demonstrated the most active compound (15ai) exerts its antiproliferative effect through induction of ferroptosis, which has recently gained considerable interest as a potential target for anticancer therapy.

Overall, the work is a wonderful showcase for modern natural product synthesis, reporting both a highly original and clever new synthetic approach and straightforward access to unique non-natural alkaloid analogues displaying very promising biological activity. As such, the work will be of great interest to a broad scientific audience. The manuscript is generally well written (although some improvements can be made, see minor comments below for some examples) and supported by excellent figures and schemes, and the SI is complete. Consequently, I enthusiastically support publication of this excellent work in Nature Communications after minor revision as noted below.

Minor comments:

- Title (and elsewhere in the manuscript: 'alkaloids/analogues' is somewhat unclear; 'alkaloids and analogues' would be better
- Abstract: '... in short steps' should be 'in few steps' or 'in a short synthetic route'.
- Fig. 1B: The 180° bond angle in the formamide looks awkward; please correct.
- Page 2: 'However, there are several concerns remained to be addressed' should be either 'However, there are several concerns that remain to be addressed' or simply 'However, several concerns remained to be addressed'
- Page 2: 'by stabilizing the generated nitrogenous radical intermediate': this is a bit confusing, as the homo-Mannich reaction is presented as a cationic process (e.g. in Fig. 1B). Please clarify/elaborate.
- Page 3: 'or its bioisosterisms' should be 'or its bioisosteres'
- Page 3: 'all the tested compounds showed no obvious cytotoxic effects'; suggest change to 'none of the tested compounds showed obvious cytotoxic effects'
- Ref. 26: Please include the title of this reference.
- Ref. 31: Please check the name of the first author.
- The procedure for the Kulinkovich cyclopropanation (SI) mentions the use of 'freshly prepared $\text{TiCl}_2(\text{OiPr})_2$ '; please include a procedure for the preparation of this reagent.

Reviewer #3:

Remarks to the Author:

Zhang and coworkers, in this paper, developed a versatile method for the diversified syntheses of Sarpagine-type monoterpene indole alkaloids and their analogues. This method features with a protecting-group-free, redox economic and short-step to construct the tetracyclic sarpagine core from Ltryptophan esters. The key step was an interesting Bischler-Napieralski/homo-Mannich sequence, which efficiently provided the indole-fused 9-azabicyclo[3.3.1]nonane sarpagine skeleton. This is a fine improvement compared to their first-generation homo-Mannich reaction of cyclopropanol with the iminium ion generated by C-H oxidation, and set a nice example for future total synthesis. As a plus, this paper deals also with the biological studies of the synthesized compounds. The results of systematic anticancer evaluation and further mechanism studies are of great interest to the community in this field. The Supporting Information is sufficient and well-organized. Overall, the present work is suitable for publication in Nature Communications.

Just a few minor revisions are required.

- (1) In Paragraph 2, the authors mainly emphasized the previous achievements on the synthesis of sarpagine-type alkaloids, which is appreciable. However, a few sentences should be given to credit the previous biological activities of sarpagine alkaloids, as it is equally important for this paper.
- (2) In Paragraph 2, a number of strategies has been listed in this paragraph, and refs 15-35 should be properly cited and organized according to the listed strategy, thus, the readers could catch the Refs conveniently.
- (3) In the Supplementary Information, S3, Line-197, the structures for compound 11 and 27 should be checked.

REVIEWER COMMENTS

Reviewer 1

Comment 1: In this manuscript, Zhang and co-workers reported a highly efficient approach to access the tetracyclic core of sarpagine-type alkaloids, hinging on a sequential Bischler-Napieralski reaction and homo-Mannich reaction as the key element. While the same group has published a series of seminal works on the subject of homo-Mannich reaction of cyclopropanol with the iminium ion, the essence of the present study is to combine the unique tactics with the classical Bischler-Napieralski reaction, which offers a more redox economic method than the previous one. Based on the newly developed methodology, a collection of sarpagine-type alkaloids and their analogues have been achieved in short steps. Moreover, the authors also evaluated the anti-cancer activity of the obtained natural products and their analogs. Encouragingly, they identified one synthetic analogue that displayed significantly improved anti-cancer activity than the natural products. More importantly, the mechanism of action of the identified active compound was also conducted, which revealed that it exerted the antiproliferation effects by serving as a promoter of ferroptosis. It should be noted that ferroptosis has recently emerged as a hot subject in the biomedical research area. In this context, the present work provides a new chemotype of ferroptosis promoter, which may serve as a new lead for the development of novel anti-cancer drugs. Collectively, the reported discoveries in this work are of great significance in both chemistry and biology. The manuscript is well organized and the supporting information also looks excellent. Thus, I would like to recommend its acceptance by Nature Communication with minor revisions.

Our response: We appreciate the highly positive comments from this knowledgeable reviewer.

Comment 2: Page 1, left column, for the reference 15-35, it is better to put the corresponding reference(s) just behind the mentioned reactions. It will help the readers to know which reference(s) correlates to which reaction.

Our Response: Thanks for this suggestion. All these references have been categorized according to the reaction type in this revised manuscript.

Comment 3: Page 2, left column, “.....to find an appropriate reaction condition for.....”, should be “.....to find appropriate reaction conditions for.....”.

Our Response: Thanks for this suggestion. The manuscript has been revised accordingly.

Comment 4: Page 2, right column, “Among these, Fe(OTf)₃ was identified.....” should be “Among them, Fe(OTf)₃ was identified.....”.

Our Response: Thanks for this suggestion. The manuscript has been revised accordingly.

Comment 5: Page 2, “Starting with 15a, a sequence including N-tosylation, hydroxymethylation with Cs₂CO₃/paraformaldehyde and detosylation produced diol 18 in high efficiency.” In this reaction sequence, a disproportionated reaction should occur after the hydroxymethylation reaction, since the aldehyde needs to be reduced to the alcohol. However, this reaction was not mentioned in the manuscript, which makes the sentence misleading. Moreover, the corresponding reference for the one-pot hydroxymethylation and disproportionated reaction should be included in this place.

Our Response: Thanks for this suggestion. The manuscript has been revised to “...Starting with 15a, a sequence including N-tosylation, domino aldol/Cannizzaro reactions by treating with Cs₂CO₃/paraformaldehyde,^{28,46} and N-detosylation...” accordingly, and a related paper (JACS 2003, 125, 4048) has been cited as ref. 46.

Comment 6: In Fig. 3, some figures look vague, which should be replaced with high-quality ones.

Our Response: Thanks for this suggestion. Clearer figures have been provided in the revised manuscript.

Comment 7: For the partial hydrogenation of the allene group in the total synthesis, how comes the excellent diastereoselectivity (cis vs trans)? Is there any rationalization for the result? If there are some references for this transformation, they should be included in the corresponding place.

Our Response: For the partial hydrogenation of the allene group, the selective hydrogenation took place from the less hindered face to afford the products 4, 5, 6, 8, and 10 with E/Z>20:1. Accordingly, related papers (Bosch, JACS, 1997, 119, 7230; MacMillan, JACS, 2013, 135, 6442) have been cited as ref. 44 and 45.

Reviewer 2

Comment 1: Zhang and co-workers report a Bischler-Napieralski/homo-Mannich sequence as a new entry to the core of the sarpagine alkaloids. This new synthetic approach is highly efficient, as 1) the starting materials (1) are accessible from the corresponding tryptophan derivatives, 2) the two key steps involve only very standard reagents and techniques, and 3) no protective groups are required. As a result, this new approach represents a superior strategy to access the valuable natural product scaffold 3. The authors were able to prepare not only no less than five complex alkaloids in very efficient sequences, but also a wide variety of non-natural analogues (15ad-ar), allowing for a systematic evaluation of the anticancer properties of this compound class.

Intriguingly, the naturally occurring alkaloids were shown to display modest proliferation inhibition at best, however, selected synthetic analogues (e.g. 15ai) showed up to ~10-fold enhanced activity. Moreover, the authors demonstrated the most active compound (15ai) exerts its antiproliferative effect through induction of ferroptosis, which has recently gained considerable interest as a potential target for anticancer therapy.

Overall, the work is a wonderful showcase for modern natural product synthesis, reporting both a highly original and clever new synthetic approach and straightforward access to unique non-natural alkaloid analogues displaying very promising biological activity. As such, the work will be of great interest to a broad scientific audience. The manuscript is generally well written (although some improvements can be made, see minor comments below for some examples) and supported by excellent figures and schemes, and the SI is complete. Consequently, I enthusiastically support publication of this excellent work in Nature Communications after minor revision as noted below.

Our response: We appreciate the highly positive comments from this knowledgeable reviewer.

Comment 2: Minor comments: Title (and elsewhere in the manuscript: 'alkaloids/analogues' is somewhat unclear; 'alkaloids and analogues' would be better.

Our Response: Thanks for this suggestion. The phrase 'alkaloids/analogues' has been replaced with 'alkaloids and analogues' through the revised manuscript including the title.

Comment 3: Abstract: '... in short steps' should be 'in few steps' or 'in a short synthetic route'.

Our Response: Thanks for this suggestion. The phrase "... in short steps" has been changed to "in a short synthetic route" in the revised manuscript.

Comment 4: Fig. 1B: The 180° bond angle in the formamide looks awkward; please correct.

Our Response: We have corrected the bond angle for the formamide unit accordingly.

Comment 5: Page 2: 'However, there are several concerns remained to be addressed' should be either 'However, there are several concerns that remain to be addressed' or simply 'However, several concerns remained to be addressed'.

Our Response: Thank you for your patience. The manuscript has been revised accordingly.

Comment 6: Page 2: 'by stabilizing the generated nitrogenous radical intermediate': this is a bit confusing, as the homo-Mannich reaction is presented as a cationic process (e.g. in Fig. 1B). Please

clarify/elaborate.

Our Response: Thanks for this suggestion. Ring-opening cyclization of cyclopropanols to imines can take place through radical or ionic mechanism depending on the nature of the reacting substrates and reaction conditions. In our CuCl₂-catalyzed homo-Mannich reaction of cyclopropanol with the in situ-formed iminium ions (ref. 17 and 18), the radical mechanism was proposed based on experimental and calculational studies. According to the literature (ref. 3–7), cyclopropanol is prone to be oxidized to radical in the presence of Fe^{III} salt, therefore, a radical mechanism was also tentatively adopted in this Fe(OTf)₃-promoted homo-Mannich reaction. Accordingly, the full arrow has been changed to half arrow in Fig. 1B, and “Cyclopropanol is prone to be oxidized by Fe^{III} salts to a β-keto radical intermediate,^{3–7} therefore, this homo-Mannich reaction was postulated to go through a radical process” has been added in the first paragraph of the results and discussion part, and related reviews (Chem. Rev. **2003**, *103*, 2597; Green Synth. Catal. **2022**, *3*, 219) have also been added as ref. 3 and ref.5.

Comment 7: Page 3: ‘or its bioisosterisms’ should be ‘or its bioisosteres’.

Our Response: Thanks for this suggestion. The manuscript has been revised accordingly.

Comment 8: Page 3: ‘all the tested compounds showed no obvious cytotoxic effects’; suggest change to ‘none of the tested compounds showed obvious cytotoxic effects’

Our Response: Thanks for this suggestion. The manuscript has been revised accordingly.

Comment 9: Ref. 26: Please include the title of this reference.

Our Response: Thanks for this suggestion. We have added the title of this reference in the revised manuscript, and the number of this reference has been updated to ref. 29.

Comment 10: Ref. 31: Please check the name of the first author.

Our Response: Thanks for this suggestion. We have checked this reference, and corrected the name of the first author. The number of this reference in the revised manuscript has been updated to ref.34.

Comment 11: The procedure for the Kulinkovich cyclopropanation (SI) mentions the use of ‘freshly prepared TiCl₂(OiPr)₂’; please include a procedure for the preparation of this reagent.

Our Response: Thanks for this suggestion. The procedure for the preparation of TiCl₂(OiPr)₂ has been provided in SI.

Reviewer 3

Comment 1: Zhang and coworkers, in this paper, developed a versatile method for the diversified syntheses of Sarpagine-type monoterpene indole alkaloids and their analogues. This method features with a protecting-group-free, redox economic and short-step to construct the tetracyclic sarpagine core from L tryptophan esters. The key step was an interesting Bischler-Napieralski/homo-Mannich sequence, which efficiently provided the indole-fused 9-azabicyclo[3.3.1]nonane sarpagine skeleton. This is a fine improvement compared to their first-generation homo-Mannich reaction of cyclopropanol with the iminium ion generated by C–H oxidation, and set a nice example for future total synthesis. As a plus, this paper deals also with the biological studies of the synthesized compounds. The results of systematic anticancer evaluation and further mechanism studies are of great interest to the community in this field. The Supporting Information is sufficient and well-organized. Overall, the present work is suitable for publication in Nature Communications. Just a few minor revisions are required.

Our response: We appreciate the highly positive comments from this knowledgeable reviewer.

Comment 2: In Paragraph 2, the authors mainly emphasized the previous achievements on the synthesis of sarpagine-type alkaloids, which is appreciable. However, a few sentences should be given to credit the previous biological activities of sarpagine alkaloids, as it is equally important for this paper.

Our response: Thanks for this suggestion. The manuscript has been revised accordingly, as “Despite these advances, only a small proportion of sarpagine alkaloids have been subjected to preliminary tests including anti-hypertension, anti-cancer, and anti-inflammation,^{19–22} and in-depth biological evaluations remained to be performed.” A related review (Molecules **2018**, *23*, 943) has been cited as ref. 21.

Comment 3: In Paragraph 2, a number of strategies has been listed in this paragraph, and refs 15–35 should be properly cited and organized according to the listed strategy, thus, the readers could catch the Refs conveniently.

Our response: Thank you for your suggestion. We have formatted all the references accordingly.

Comment 4: In the Supplementary Information, S3, Line-197, the structures for compound 11 and 27 should be checked.

Our response: Thanks for this suggestion. The structures for compounds 11 and 27 have been corrected.

Reviewers' Comments:

Reviewer #2:

Remarks to the Author:

In their revision, the authors have carefully addressed the comments of all reviewers. This fine work is now suitable for publication in Nature Communications.

REVIEWER COMMENTS

Reviewer #2 (Remarks to the Author):

In their revision, the authors have carefully addressed the comments of all reviewers. This fine work is now suitable for publication in Nature Communications.

Our response: We appreciate the highly positive comments from this knowledgeable reviewer.